# Testing the Contribution of Multi-Source Remote Sensing Features for Random Forest Classification of the Greater Amanzule Tropical Peatland

**DOI:** 10.3390/s21103399

**Published:** 2021-05-13

**Authors:** Alex O. Amoakoh, Paul Aplin, Kwame T. Awuah, Irene Delgado-Fernandez, Cherith Moses, Carolina Peña Alonso, Stephen Kankam, Justice C. Mensah

**Affiliations:** 1Department of Geography and Geology, Edge Hill University, Ormskirk L39 4QP, UK; amoakoha@edgehill.ac.uk (A.O.A.); awuahk@edgehill.ac.uk (K.T.A.); delgadoi@edgehill.ac.uk (I.D.-F.); Mosesc@edgehill.ac.uk (C.M.); 2Grupo de Geografía Física y Medio Ambiente, Department of Geography, University of Las Palmas de Gran Canaria, 35003 Las Palmas, Spain; carolina.pena@ulpgc.es; 3Hen Mpoano (Our Coast), Takoradi WS-289-9503, Ghana; skankam@henmpoano.org (S.K.); jmensah@henmpoano.org (J.C.M.)

**Keywords:** tropical peatland, random forest, feature selection, classification, Sentinel, Google Earth Engine

## Abstract

Tropical peatlands such as Ghana’s Greater Amanzule peatland are highly valuable ecosystems and under great pressure from anthropogenic land use activities. Accurate measurement of their occurrence and extent is required to facilitate sustainable management. A key challenge, however, is the high cloud cover in the tropics that limits optical remote sensing data acquisition. In this work we combine optical imagery with radar and elevation data to optimise land cover classification for the Greater Amanzule tropical peatland. Sentinel-2, Sentinel-1 and Shuttle Radar Topography Mission (SRTM) imagery were acquired and integrated to drive a machine learning land cover classification using a random forest classifier. Recursive feature elimination was used to optimize high-dimensional and correlated feature space and determine the optimal features for the classification. Six datasets were compared, comprising different combinations of optical, radar and elevation features. Results showed that the best overall accuracy (OA) was found for the integrated Sentinel-2, Sentinel-1 and SRTM dataset (S2+S1+DEM), significantly outperforming all the other classifications with an OA of 94%. Assessment of the sensitivity of land cover classes to image features indicated that elevation and the original Sentinel-1 bands contributed the most to separating tropical peatlands from other land cover types. The integration of more features and the removal of redundant features systematically increased classification accuracy. We estimate Ghana’s Greater Amanzule peatland covers 60,187 ha. Our proposed methodological framework contributes a robust workflow for accurate and detailed landscape-scale monitoring of tropical peatlands, while our findings provide timely information critical for the sustainable management of the Greater Amanzule peatland.

## 1. Introduction

Peatlands are characterized by the accumulation of partially decayed organic matter formed from plant debris under waterlogged conditions [1]. They provide a wide range of ecosystem services including carbon sequestration/storage, climate change mitigation, improvement of water quality and runoff regulation, and the provision of a landscape with cultural, recreational and livelihood values. Globally, peatlands hold an estimated 650 billion tonnes of carbon on 3% of the Earth’s land surface, the equivalent to more than half of the carbon in the atmosphere or the carbon stored by Earth’s vegetation [2]. For their multiple benefits, the need for peatland conservation is widely recognized (i.e., the United Nations Framework Convention on Climate Change, Ramsar Convention on wetlands, the Convention on Biological Diversity, United Nations Convention to Combat Desertification) but has been hampered by short term economic priorities and national development policies [3,4,5]. Large areas of peatlands have already been degraded (estimated 20–25%), and remaining areas are quickly disappearing as a result of logging and plantation development, conversion to residential and industrial zones, climate change impacts and accidental burning [2,6,7,8,9]. Not only do these land use changes reduce biodiversity, they also turn peatlands into net emission sources of greenhouse gases (GHGs) at a faster rate since draining of peatlands release greenhouse gases such as carbon dioxide from the carbon stored within peat soils. 

For sustainable management of remaining peatlands, a better understanding of fundamental variables including their spatial distribution and extent is required. However, considerable uncertainties about these variables remain, particularly in the tropics. Generally, peatland research has focused strongly on boreal and temperate peatlands, with tropical peatlands receiving much less attention. The past two decades have seen increased interest in tropical peatland research [10,11,12], but most studies have focused on Southeast Asia where an estimated 56% of tropical peatlands exist [13]. By comparison, African peatlands are wholly understudied. Indeed, there remains a basic uncertainty about the existence–extent and distribution–of peatlands in Africa. For instance, a very large peatland, approximately 14,550,000 ha in area and storing an estimated 30.6 billion tonnes of carbon, was discovered only recently in the Congo Basin [14]. In Ghana, the Greater Amanzule landscape has been reported as a tropical biodiversity hotspot undergoing rapid development from agricultural plantation and urbanization [15,16]. Also, with an influx of oil and gas activities on the Greater Amanzule landscape from 2011, a complex array of pressures for peatland conversion are expected to intensify over at least the next decade [16]. However, to date, the spatial distribution and extent of peatland on the landscape is not fully known. Generally, there is limited research on practical and cost-effective remote sensing application (e.g., analysis on the sensitivity of tropical peatland to image features) for tropical peatland mapping. Developing detailed, comparable and robust tropical peatland maps in Africa is therefore an urgent priority–to inform policymakers and conservation practitioners and to provide critical information for landscape planning and for enhancing authorities’ capacity in monitoring, reporting, and verification (MRV).

Mapping allows both the location and quantification of the extent of pristine peatlands, as well as identification of areas at risk due to their proximity to degraded zones [17]. In addition, mapping functioning peatland areas can provide data to inform spatially explicit and realistic restoration and protection goals. However, landscape mapping of tropical peatlands remains a challenge, especially at regional, national and global scales, and this has resulted in their consistent under-representation, or complete omission from, many global vegetation maps [18,19].

Methodologically, optical data from high spatial resolution sensors such as Sentinel-2 (10 m multispectral imagery) and Landsat (30 m multispectral imagery) have been the primary and most successful tool for mapping peatlands [20,21,22,23,24,25,26] mainly due to their spectral detail. The enhanced spectral capabilities of optical sensors help to derive numerous band ratios and indices, such as spectral vegetation indices (VIs) for monitoring vegetation species [27,28,29]. VIs have several advantages over stand-alone spectral bands, including decreased effect of soil background on canopy reflectance, enhanced variability of spectral reflectance of target vegetation, reduced effect of atmospheric conditions and canopy geometry, and shading [30,31]. The application of optical sensing is however constrained by the frequent cloud coverage in the tropics. For monitoring high cloud coverage areas, some developments have been reported in using radar products [32,33,34]. Radar can penetrate cloud cover and is also sensitive to variable soil moisture conditions which makes it suitable for wetland mapping [19]. It offers detailed information on the often difficult to detect characteristics of vegetation such as moisture, roughness and shape [35]. Additionally, data from the Shuttle Radar Topographic Mission (SRTM) have been used to successfully identify hydrological landscape units in cloud-persistent areas (e.g., [36,37,38]). SRTM data provides an estimate of elevation and is useful for identifying large-scale topographical boundaries within tropical landscapes. 

Recent classification approaches favour the integration of data from multiple sensors for improved landscape characterization (e.g., [36,39,40,41]). Because of their complementarity, optical, radar and topographical data fusion presents an increased opportunity to map peatlands at fine scales in equatorial zones affected by cloud cover, although the choice of appropriate features from these datasets remains a challenge. Past efforts to partially map the Greater Amanzule peatland (predominantly mangrove) have relied on a single source of data [42,43], plus participatory GIS and ground referencing methods [15,16]. In the present study, optical, radar and elevation remote sensing data and their various combinations are used to classify the entire Greater Amanzule landscape. We determine the optimal classification approach for the Greater Amanzule tropical peatland based on the integration of image features derived from Sentinel-2, Sentinel-1 and SRTM data–thus to advance geospatial methodologies for mapping tropical peatland. Specifically, the study (i) defines the extent and distribution of peatland on the Greater Amanzule landscape; (ii) proposes a framework for extracting appropriate Sentinel-2, Sentinel-1 and SRTM image features for tropical peatland mapping; (iii) assesses how classifications of different combinations of Sentinel-2, Sentinel-1 and SRTM image features compare; and (iv) determines the sensitivity of land cover types to multi-source data features. 

## 2. Materials and Methods

### 2.1. Study Area

The Greater Amanzule tropical peatland is located in the Western Region of Ghana. The Region is bordered to the west by Cote D’Ivoire, to the east by the Central Region, to the north by Ashanti and Western North regions and to the south by the Gulf of Guinea [16]. The catchment of the peatland lies within the Wet Evergreen Forest zone of Ghana and traverses the four coastal Districts of Jomoro, Ellembele, Ahanta West and Nzema East (Figure 1). The Greater Amanzule system is made up of a relatively pristine wetland complex consisting of freshwater lagoons, rivers, forests, and grasslands [15]. The peatland is patchy and access to its resources is uneven for fringe communities, influenced by factors such as proximity and characteristics of the peat resources [15]. The area is rich in indigenous avifauna and hosts various migrant species. It is classified as an Important Bird and Biodiversity Area (IBA) by Birdlife International and it meets the criteria for designation as a Wetland of International Importance according to the Ramsar Convention [44]. Another important biodiversity characteristic of the site is the coastal area constituting important turtle nesting sites [44]. Species include leatherback (*Dermochelys coreacea*), green (*Chelonia mydas*), olive ridley (*Lepidochelys olivacea*) and hawksbill (*Eretmochelys imbricate*) turtles.

The study area falls within the equatorial climate zone, which is characterized by moderate temperatures (annual average of 26 °C), and high rainfall and relative humidity (annual averages of 1600 mm and 87.5%, respectively [45]). It experiences a double-maxima pattern of rainfall with peaks in May–June and October–November, and a short dry season (December–March) during which north-westerly winds bring slight harmattan conditions. The hydrology of the area is driven by six rivers, one lake, and four estuaries [45]. 

The coastline where the study site is located is comprised of regular sandy beaches with no headlands or rocky outcrops. The hinterland is generally low-lying and relatively flat (Figure 2). These low-lying coastal areas extend inland after which the topography becomes hilly.

Characteristic soils are predominantly forest oxysols [45]. The estimated mean aboveground carbon stocks of swamp forest are 1.1429 × 10^−7^ ± 1.592 × 10^−8^ tC/ha in intact areas, 7.127 × 10^−8^ ± 1.026 × 10^−8^ tC/ha in degraded areas, and 2.046 × 10^−8^ ± 1.212 × 10^−8^ tC/ha in deforested areas. The belowground carbon stocks for intact swamp forest is estimated at 2.286 × 10^−8^ ± 3.18 × 10^−9^ tC/ha, for degraded areas at 1.425 × 10^−8^ ± 2.05 × 10^−9^ tC/ha, and for deforested areas at 4.09 × 10^−9^ ± 2.43 × 10^−9^ tC/ha [45]. Ajonina et al. [46] also estimated the total above ground carbon stored in intact mangrove forest at 65–422 tC/ha with mean of 185 tC/ha.

In terms of socio-economic activities, local gin brewing, charcoal production, small-scale farming and fishing, cash crop farming (rubber, coconut and palm oil) and small scale trading are dominant. The major threats to the Greater Amanzule peatland include illegal alluvial gold mining activities, the spread of the giant aquatic grass *Vossia Cuspidata* resulting from eutrophication of the waters, which is affecting fishing, the absence of a formal management regime for the landscape and the subsequent breakdown of traditional conservation approaches to resource management. 

Other threats include capture and consumption of turtles, bush burning, illegal logging, charcoal burning, influx of oil and gas activities on the landscape, expansion of plantations, illegal hunting of wildlife, and mangrove harvesting for fish smoking. Colonization by the fern *Acostricum* of mangrove habitats is another concern [15,16].

### 2.2. Satellite Remote Sensing Data

Multi-source and multitemporal optical and radar data were combined for land cover classification of the Greater Amanzule landscape (Figure 2). A time series of multispectral Sentinel-2 and Sentinel-1 imagery was collected throughout 2019, providing a data stack of the whole annual cycle. Sentinel data provided by the European Space Agency (ESA) are publicly available and have a relatively fine spatial resolution of 10–60 m for Sentinel-2 and 10 m for Sentinel-1. The Sentinel-2 instrument has 13 spectral bands covering the visible and near infrared portions of the electromagnetic spectrum, including four red-edge bands (703.9 nm–864.8 nm) which were traditionally only available to hyperspectral sensors and have the advantage of providing key information on the state of vegetation. The Sentinel-1 dual-polarization C-band Synthetic Aperture Radar (SAR) data originated from the Level-1 Ground Range Detected (GRD) Interferometric Wide Swath (IW) products as ingested in Google Earth Engine (GEE) [47]. The frequent revisit periods of the Sentinel constellations (5 days for Sentinel-2 and 6-12 days for Sentinel-1) enabled large collections of images: 366 Sentinel-2 images and 63 Sentinel-1images (Table 1). This made it possible to create a high-quality composite based on the average of many pixel scenes, thus reducing atmospheric and seasonal effects on classification. Further, elevation data from SRTM at 30 m resolution was incorporated as ancillary data. These input data—Sentinel-2, Sentinel-1 and SRTM—were arranged into six data combinations (described below in feature extraction and selection section) for comparative land cover classification analysis.

Pre-processing of the data was carried out separately for the Sentinel-2 and Sentinel-1 datasets. Cloudy pixels in the Sentinel-2 imagery were eliminated using the cloud mask (QA60 band) provided with Sentinel-2 data as well as a modified Landsat cloudScore algorithm built to detect clouds using bands B1, B2, B8, B10 and B11 [41,48]. The algorithm was customized for the region to create cloud-free image composites. The final Sentinel-2 image was produced by computing the mean of all bands in the 40 to 60 percentile range [41]. Compared to the full range of values, the 40–60 percentile range is less variable, temporally stable and captures a more representative spectral characterization of land cover by minimizing extreme atmospheric effects—given that we used a Level 1C product—and the challenge of cloud persistence. Further, this helps to minimize errors due to seasonal and phenological variations.

Sentinel-1 GRD products available in GEE had been subjected to the following processing steps using the Sentinel-1 Toolbox: thermal noise removal, radiometric calibration, and terrain correction, to generate a calibrated and ortho-corrected product. Two different polarisation bands were selected: single co-polarisation with vertical transmit/receive (VV) and dual-band co-polarisation with vertical transmit and horizontal receive (VH). 

The fused datasets were resampled to 20 m spatial resolution for the classification using the nearest neighbour resampling method. The 20 m resolution was to ensure consistency with the ‘red edge’ bands (B5, B6, B7, B8A) in Sentinel-2 which are useful in classifying visually similar tree crops [49]. 

### 2.3. Reference Data and Classification Scheme

Reference data were collected for algorithm parameterization and training as well as accuracy assessment of land cover classification. Reference data were acquired from several sources, including existing inventory maps [15,16] and field surveys based on a stratified random sampling approach to capture spatial and spectral landscape variations while minimizing autocorrelation. Field survey was carried out by three of the research team with extensive experience and familiarity with the landscape. Field survey data were cross-referenced with interpretation of very high-resolution imagery available in Google Earth Pro. The training data were delineated as polygon features and separate testing data delineated as point features. The polygon extents were then used to extract image pixels to train the classifier. Using polygon features helps to capture sufficient spectral variation for each land cover class and has been found to produce better classification outcomes relative to other approaches such as points (single pixels), point buffers (average pixel value) and image objects (area statistics) [50,51]. A total of 50,581 pixels were collected as reference data (Table 2). 

Classifying large tropical landscapes from remote sensing data is complex due to structural complexity, high heterogeneity and the absence of a universal classification scheme for peatlands. Defining appropriate thematic classes based on the characteristics of the study area and technical specifications of the imagery is therefore important. A classification system was adopted based on extensive field observations of the study area and a careful study of relevant literature. Peatland classes were adapted from Lawson et al. [19] and comprised mangrove swamp, mixed swamp, palm swamp and bog plain; plantation comprised coconut, rubber and oil palm; artificial and bare classes included built-up land and bare surfaces, respectively. Other classes included sparse vegetation, natural forest and water. In total, twelve classes were identified (described in Table 2).

### 2.4. Feature Extraction and Selection

Feature extraction is the derivation of new features from original image bands, e.g., deriving vegetation indices from Sentinel-2 bands and texture features from Sentinel-1 bands. Feature selection is the removal of irrelevant or redundant features, either original or derived, from complete datasets [32]. This process was undertaken to create different data combinations for integrated classification and subsequent comparison. All Sentinel-2, Sentinel-1 and SRTM data features were derived using GEE. A total of 36 features reported to be effective for vegetation and other land cover delineation were considered in our study (Table 3). These included 12 original bands, 10 VIs, nine texture features, two temporal features and three elevation features. Classification was conducted using six datasets: (1) We first tested the original Sentinel-2 image bands (S2) since optical imagery is the standard data source used for peatland classification. (2) Next, we extracted further spectral features—principally VIs—from the Sentinel-2 bands to create an enhanced Sentinel-2 dataset (S2+). (3) We then tested the original Sentinel-1 bands (S1) to assess the contribution of radar data for peatland classification, even though it was expected that this dataset would prove limited on its own. (4) Next, we extracted further texture and temporal features from the Sentinel-1 bands to create an enhanced Sentinel-1 dataset (S1+). (5) Then we combined the enhanced Sentinel-2 and Sentinel-1 datasets (S2+S1+) to assess an integrated optical-radar dataset for classification. (6) Finally, to this combined Sentinel-2 and Sentinel-1 dataset, we added STRM-derived elevation features to assess an integrated optical-radar-DEM dataset for classification (S2+S1+DEM). The feature combinations used for the classification analysis are presented in Table 3.

#### 2.4.1. Vegetation Indices

Vegetation indices (VIs) are parameters sensitive to photosynthetic active radiation and are commonly computed from the spectral reflectance of two or more bands. Ten indices were calculated from the Sentinel-2 image: the normalized difference vegetation index (NDVI; Equation (1)), the normalized difference water index (NDWI; Equation (2)), land surface water index (LSWI; Equation (3)), enhanced vegetation index (EVI; Equation (4)), atmospherically resistant vegetation index (ARVI; Equation (5)), normalized burn ratio (NBR; Equation (6)), normalized burn ratio 2 (NBR2; Equation (7)), green normalized difference vegetation index (GNDVI; Equation (8)), Sentinel-2 red-edge position index (S2REP; Equation (9)), and modified soil-adjusted vegetation index (MSAVI2; Equation (10)). All the indices were calculated at 20 m resolution using the equations below;
(1)$$\mathrm{NDVI}=\frac{\rho \mathrm{NIR}-\rho \mathrm{Red}}{\rho \mathrm{NIR}+\rho \mathrm{Red}},$$
(2)$$\mathrm{NDWI}=\frac{\rho G\mathrm{reen}-\rho \mathrm{NIR}}{\rho \mathrm{Green}+\rho N\mathrm{IR}},$$
(3)$$\mathrm{LSWI}=\frac{\rho \mathrm{NIR}-\rho \mathrm{SWIR}1}{\rho \mathrm{NIR}+\rho \mathrm{SWIR}1},$$
(4)$$\mathrm{EVI}=2.5*\;\frac{\rho \mathrm{NIR}-\rho \mathrm{Red}}{\rho \mathrm{NIR}+6\;*\;\rho \mathrm{Red}-7.5*\;\rho \mathrm{Blue}+1},$$
(5)$$\mathrm{ARVI}=\frac{\rho \mathrm{NIR}-\left(2\rho \mathrm{Red}-\rho \mathrm{Blue}\right)}{\rho \mathrm{NIR}+\left(2\rho \mathrm{Red}-\rho \mathrm{Blue}\right)},$$
(6)$$\mathrm{NBR}=\frac{\rho \mathrm{NIR}-\rho S\mathrm{WIR}2}{\rho \mathrm{NIR}+\rho \mathrm{SWIR}2},$$
(7)$$\mathrm{NBR}2=\frac{\rho \mathrm{SWIR}1-\rho \mathrm{SWIR}2}{\rho \mathrm{SWIR}1+\rho \mathrm{SWIR}2},$$
(8)$$\mathrm{GNDVI}=\frac{\rho \mathrm{NIR}-\rho \mathrm{Green}}{\rho \mathrm{NIR}+\rho \mathrm{Green}},$$
(9)$$S2\mathrm{REP}=705+35*\left[\;\frac{\left(\frac{\rho \mathrm{RedEdge}3+\rho \mathrm{Red}}{2}\right)-\rho \mathrm{RedEdge}1}{\rho \mathrm{RedEdge}2-\rho \mathrm{RedEdge}1}\right],$$
(10)$$\mathrm{MSAVI}2=\frac{\left(2*\rho \mathrm{NIR}+1-\sqrt{{\left(2*\rho \mathrm{NIR}+1\right)}^{2}-8*(\rho \mathrm{NIR}-\rho \mathrm{Red}}\right))}{2},$$
where ρ_Blue_, ρ_Red_, ρ_Green_, ρ_NIR_, ρ_SWIR1_ and ρ_SWIR2_ are the reflectance values of the respective bands in Sentinel-2 sensor. The importance of these indices for vegetation discrimination has been widely reported [55,56,57]. For example, combining NDVI, EVI, LSWI and NBR2 has been shown to provide a better separation of a dynamic tropical peatland [39]. Another study has found that NBR, NDVI, ARVI, and LSWI are key features for discriminating between rubber plantation and other vegetation types [54]. Several studies have also demonstrated the usefulness of GNDVI, S2REP, and NDWI for mapping wetland, plantation, and other land cover types [41,48,49].

#### 2.4.2. Texture Features

Standard deviation metrics were computed on Sentinel-2 based NDVI and Sentinel-1’s VH and VV bands using a 5 × 5 pixels moving window [41]. Thus, for each central pixel in the 5 × 5 window, the standard deviation of the 25 pixels (in the window) was calculated and the value applied to the corresponding (central) pixel in the output texture image. Other radar derived texture features reported to be effective in discriminating vegetation and other land cover types were calculated using the grey-level co-occurrence matrix (GLCM) [49,58,59]; the size of the neighbourhood to include in each GLCM was set to 4 and the kernel was a 3 × 3 square using the glcmTexture function in GEE. Kernel sizes in part determine the success of texture-based image classification. If the window size is too small, enough spatial information cannot be extracted to distinguish among different land features. If the window size is too large, it could overlap different features and introduce spatial errors [60]. Our choice of kernel size followed literature in similar contexts [39,41]. The formulae of indicators used are shown in Equations (11)–(13):(11)$$\mathrm{Contrast}=\overset{N_{g}-1}{\underset{i,j=0}{\sum }}{\left(i-j\right)}^{2}\mathrm{GLCM}\;\left(i,j\right),$$
(12)$$\mathrm{Correlation}=\overset{N_{g}-1}{\underset{i,j=0}{\sum }}GLCM\left(i,j\right)\left[\;\frac{\left(i-\mu_{i}\right)\left(j-\mu_{j}\right)}{\sigma_{i}\;\times \;\sigma_{j}}\right],$$
(13)$$\mathrm{Variance}\;=\overset{N_{g}-1}{\underset{i,j=0}{\sum }}\mathrm{GLCM}\left(i,j\right){\left(i-\mu \right)}^{2}$$
where GLCM (*i*,*j*) is the entry in a normalized grey-level co-occurrence matrix; $N_{g}$ is number of distinct image grey levels, and $\mu_{i}$, $\mu_{j}$ and $\sigma_{i},\;\sigma_{j}$ are the mean and standard deviations respectively. Contrast describes the degree of chromatic change between neighbouring pixels; correlation measures the linear dependencies of grey levels for neighbouring pixels; and variance measures dispersion between neighbouring pixels; all of which were derived from GLCM [39].

#### 2.4.3. Temporal Features

Temporal features depicting soil moisture conditions were derived by computing the amplitude decrease between SAR images (i.e., amplitude change between the 10th and 90th percentiles). This approach has already been shown suitable for relative surface soil moisture retrieval from SAR [61,62,63].

#### 2.4.4. Elevation Features

Lidzhegu et al. ([38], p. 97) defined a wetland as “*an environment where fluvial and/or tectonic processes have shaped the landscape such that topographic conditions become suited for prolonged inundation sufficient for soil oxidation and establishment of hydrophytes*”. Inherent in this definition is the importance of topographic features for wetland delineation. Tropical wetlands occur as linear features corresponding to the alignment of valley bottoms. Ancillary features that help to identify topographic effects on hydrological processes were thus derived from the SRTM digital elevation data at 30 m resolution [64]. A total of three features were derived at each DEM pixel: slope, aspect and elevation [14,36,38].

### 2.5. Classification and Accuracy Assessement 

The image features were standardized by subtracting the mean and scaling to unit variance prior to classification. To deal with the challenge of overfitting, the recursive feature elimination (RFE) algorithm was used to reduce the number of features by eliminating the least important features based on stratified 2-fold cross-validation scores. Important features were then used to retrain a random forest (RF) model for the classification. The RF machine learning is an ensemble classifier that combines decision trees, bootstrap aggregation (bagging) and random subspace methods for classification and regression [65]. The combination of many weak learners in an ensemble thus contributes to RF achieving higher accuracy compared to machine learning algorithms based on a single classifier [65,66]. RF has become increasingly important in land cover classification in recent times because of its nonparametric nature, ability to limit overfitting and its flexibility [32,53]. It is also known for its high performance and efficiency in dealing with large input datasets with many different features [65]. A machine learning analysis to assess the classification capabilities of 179 classifiers revealed RF as the best classifier among others that included support vector machine (SVM), decision trees, and neutral networks [67]. In their analysis of four classifiers, Kaszta et al. [68] and Awuah et al. [51] also identified RF and SVM as the best performers among others that included k-nearest neighbours (kNN) and classification and regression trees (CART). 

Two important parameters need to be set up in a RF classifier: the number of trees (ntree) and the number of splits (mtry). The ntree and mtry were set at 100 and 2 respectively. Nomura et al. [41] found that accuracy did not increase beyond 100 trees. We were also interested in feature importance scores, to determine the relative contribution of each image feature in the classification of the different land cover types. Some of the most frequent approximations for feature selection in decision trees include Gini Index [69], gain-ratio [70], and Chi-square [71]. We estimated the feature importance scores for the overall classification using a RF-based Gini criterion. A RF usually uses the Gini Index as a measure for the best split selection, which measures the impurity of a given element with respect to the rest of the classes. For example, when assigning an input pixel to a class (Ci), for a given training set (T), the Gini Index measures feature impurity with respect to the different classes and is expressed as;

(14)$$\sum \underset{i\neq j}{\sum }\left(\frac{f\left(C_{i},T\right)}{\left|T\right|}\right)\left(\frac{f\left(C_{j},T\right)}{\left|T\right|}\right)\;$$
where (*f* (*C_i_*, *T*)/|*T*|) is the probability that the selected pixel belongs to class $C_{i}$ [51,72]. Feature importance scores for each land cover class were subsequently estimated from the product of the overall feature importance estimates and the standardized mean value of each feature split for the given class. 

A post-classification majority filter was applied to improve class homogeneity using a 3 × 3 pixel moving window. The accuracy of the map was evaluated using a confusion matrix [73] from which producer’s, user’s, and overall accuracies were calculated to affirm the validity and accuracy of the results. We also calculated the F-score of each land cover class for all the image stacks to determine the degree of discrimination of a given land cover class using Equation (15):(15)$$F-\mathrm{score}=\frac{2\left(\mathrm{UA}-\mathrm{PA}\right)}{\mathrm{UA}+\mathrm{PA}}$$
where UA and PA represent the user and producer accuracies of a particular land cover class. The statistical significance of the comparative accuracies of the various data combinations were also calculated using McNemar’s chi-squared test score from Equation (16);
(16)$$z=\frac{f_{12}-f_{21}}{\sqrt{f_{12}+f_{21}}}$$
where f_12_ represents the number of samples correctly classified in the first classification, but incorrectly in the second classification, and f_21_ represents the number of samples correctly classified in the second classification but incorrectly in the first. The result was assessed at α = 0.05 significance level with a z = 1.96 critical value. 

The methodological workflow of the study is summarised in the flowchart presented in Figure 3. 

## 3. Results

### 3.1. Selection of Optimal Feature Variables

After executing the RFE, the optimal number of features was obtained, as shown in Figure 4. This represents subsets of feature variables that combine to produce the best classification accuracy. The results show that all features considered for the classifications of S2, S1, and S1+ were relevant, thus, the highest accuracy for these datasets was obtained at the point where all features were used for the classification. For the other three datasets—S2+, S2+S1+, S2+S1+DEM—cross-validation scores increased significantly in the early stages and peaked when the number of features was 20, 25 and 29, respectively. Beyond the maxima for these three datasets, cross validation scores fluctuated with increasing feature numbers, signifying the presence of irrelevant or redundant features which were not increasing classification accuracy. The analysis revealed that the S2+, S2+S1+ and S2+S1+DEM datasets contained one, eight and seven redundant features respectively which needed to be removed. The relevant features of the S2+ dataset included nine original bands, 10 vegetation features and one texture feature. For the S2+S1+ dataset, relevant features included eight original bands, 10 vegetation features, six texture features and one temporal feature. For S2+S1+DEM, relevant features included 10 original bands, 10 vegetation features, six texture features, two temporal features and one elevation feature. Our analysis showed that the cross-validation score increased with increased number of image features. Accuracy scores followed a similar pattern where S1 < S1+ < S2 < S2+ < S2+S1+ < S2+S1+DEM. This shows that the presence of more relevant feature variables can result in improved classification accuracy. Table 4 shows the optimal features retained for classification. 

### 3.2. Land Cover Classification Accuracy

Sections of the six land cover classifications (S2, S2+, S1, S1+, S2+S1+, S2+S1+DEM) are presented in Figure 5. Maps were visually similar for S2 and S2+, S1 and S1+, as well as S2+S1+ and S2+S1+DEM, the differences of which were only revealed in quantitative accuracy assessment.

In general, land cover classifications were relatively accurate, with all overall accuracies approaching or exceeding 90%, except for S1 and S1+, as expected (Table 5). Of the different datasets, S2+S1+DEM produced the highest overall accuracy (94%), followed by S2+S1+ (92%) and S2+ (91%). Full error matrices of all classifications are available as Appendix A (Appendix A).

Despite the visual similarities of the classified images, a McNemar test of significance showed that accuracies of all datasets were significantly different from each other (Table 5). This finding reaffirms the contention that accurate land cover mapping requires the use of relevant features, and here feature optimization holds considerable value for the mapping community.

The UA and PA results are presented in Table 6. The S2+S1+DEM stands out with generally better results. The worst were found for S1 and S1+. To investigate the superiority of the S2+S1+DEM classification against the other datasets, we compared class UAs and PAs (Table 6). A total of 101 table cells (84%) showed improvement in either PA or UA with S2+S1+DEM compared to the other datasets, whereas 8 cells (7%) showed another dataset was better than S2+S1+DEM. This demonstrates the robust nature of the S2+S1+DEM for discriminating different land cover classes–it performed better when compared to the other datasets. For land cover classes in which the UA or PA did not improve by at least 10% over the other datasets, accuracies were already generally high (>70%). 

F-scores from all datasets are presented in Figure 6. Apart from S1 and S1+, all datasets achieved high accuracies in differentiating the peatland classes (mangrove swamp, mixed swamp, palm swamp and bog plain), with an F-score between 0.91 and 0.99. S2+S1+DEM had the best F-score for all the land cover classes, between 0.80 and 0.99, which indicated that this approach has strong potential for land cover classification, particularly for tropical peatland mapping. The results also reaffirm the ability of the RF machine learning algorithm to map complex landscapes accurately [32,65,66,67,68]. 

### 3.3. Feature Importance

A summary of the most important features for each dataset is presented in Figure 7. The results varied considerably, depending on the feature types used in training the RF classifier. 

When the classifier was trained with the full dataset (S2+S1+DEM), elevation, was the most important predictor variable, thereby highlighting the important role of topography in peatland delineation. Although the S1 and S1+ datasets were relatively ineffective for classification on their own, the radar derived features were consistently important predictor variables in multi-source datasets, VH, VV and VH standard deviation in particular. Red Edge 2 was the most important variable for the S2 dataset, but was deemed irrelevant in S2+, S2+S1+, and S2+S1+DEM, showing that optimizing feature sets by removing irrelevant features is an important step to avoid assumptions that can lead to reduced classification accuracy. From this point on, we focus on S2+S1+DEM, owing to its general superiority over the other datasets. 

The sensitivity of the 12 land cover classes to the 36 features of the S2+S1+DEM (described in Table 3 in Section 2.4) is demonstrated in Table 7. The original Sentinel-1 bands (VV and VH) were consistently important predictors for most of the vegetation classes (both peatland and non-peatland vegetation classes) on the landscape. Elevation feature derived from the SRTM data was also very important for discriminating vegetation types, indicating that the spatial distribution of vegetation types is greatly influenced by topographic information. One image feature which also stood out as being very important for differentiating unvegetated from vegetated areas is Sentinel-2’s SWIR 2 band. Results on how the other datasets discriminated the various land covers are presented as Appendix A (Appendix A).

### 3.4. Classification of the Greater Amanzule Tropical Peatland Using the S2+S1+DEM Dataset

The extent and distribution of the Greater Amanzule landscape are presented in Table 8 and Figure 8, respectively. Water aside, peatland classes constituted a significant proportion of the landscape (23% of total area without water)—dominated by mixed swamp, palm swamp, bog plains, and mangroves, respectively. The result clearly demonstrates a largely vegetated landscape, with patches of built-up land making up only 5273 ha (˂1%) of the total landscape. While this may suggest a largely undeveloped landscape, it is important to note that about 50,713 ha (8.7% total area) of the vegetated areas are plantations of coconut, rubber and oil palm—this represents land use conversion from natural forest and/or peatland. In the absence of clearly defined boundaries and management strategies, the plantation development in Greater Amanzule should be an environmental concern since similar land conversion has proved to be a major threat to tropical peatlands in South East Asia [74,75,76]. 

Patches of the landscape showed bare surfaces, mostly comprised of land cleared for development. Field observations showed road construction works and the development of oil and gas industries. Although limited in number, roads have already been constructed across some of the peatland of the Greater Amanzule (e.g., between Ellonyi and Kengen, and between Alabokazo and Sanzule, on the Ankobra River [15]). No studies have yet been conducted to ascertain the specific impact of roads on the peatland vegetation, hydrology or soil properties, although observation of palm swamp and mangrove die back at Sanzule and Kamgbunli respectively, during the Alabokazu-Sanzule road construction [15], suggests that roads could be having a negative impact on the peatland. Roads can divert or impede water, act as a barrier to groundwater and channel flow, and eventually lead to the degradation of peatland vegetation or carbon cycling. The siting of large oil and gas facilities in the peatland [15] may have a negative impact, if not controlled. This has the potential to drive up population, increasing demand for land and eventually peatland, and construction work may have a negative impact on the hydrology of the area.

We estimate that the total peatland area in 2019 is 60,187.04 ha. Within the landscape, mangrove occurred in patches and predominated around the Ankobra River, Bakanta, and Miemia. A large block of mixed swamp could be found along the border of Ghana and Cote D’Ivoire, on the River Tano, extending onto the Aby lagoon. Another large block of mixed swamp was found around the stilt village, Nzulezo. Palm swamp and bog plain occurred in patches, with the highest concentration around Nzulezu. These are areas currently without any formal management regime in place and will thus require effort from stakeholders, particularly local communities and government agencies, to ensure their conservation. 

In terms of the distribution of other land cover classes, coconut was highly concentrated in the western part of the landscape while rubber was highly concentrated in the east. The concentration of oil palm was high at the Cote D’Ivoire boundary of the landscape in the west, and sparse vegetation surrounded built-up areas. 

## 4. Discussion

In this study, effort was focused on developing a robust framework for mapping the Greater Amanzule tropical peatland of Ghana, using multi-source satellite imagery and a RF algorithm within the GEE environment. The proposed framework provides a systematic technique for extracting appropriate feature variables for tropical peatland classification. This has been developed by integrating original spectral bands, plus derived vegetation indices, texture and temporal features, as well as ancillary elevation data features into a single composite dataset. Multi-sensor satellite imagery has complementary characteristics which enabled improved detection of peatland and non-peatland classes. These land cover types are difficult to map using single source datasets due to structural complexity and high heterogeneity. Our results are consistent with earlier studies that combined optical and radar dataset for land cover mapping, showing that the combination produces higher overall accuracy over individual sensor dataset (e.g., [36,49,59]). The introduction of the elevation data improved the accuracy of the optical-radar data combination significantly (Table 4), thus confirming that classification enhancement may occur when a primary dataset such as SRTM is integrated with other datasets for peatland classification [77]. The lowest overall accuracy was observed with the Sentinel-1 only products (S1, S1+); this is also consistent with previous studies (e.g., [39,59,78]). Despite the cloud penetration advantage of Sentinel-1 data, it failed on its own to distinguish various land cover classes. This suggests that the best way to maximize the utility of such data for various land cover classes discrimination is to combine it with optical datasets as demonstrated in this study. Da Silva *et al.* [79] also suggested advanced techniques such as SAR polarimetry to optimize SAR for the discrimination of diverse land cover classes.

The overall accuracy of the S1 dataset (70%) reported in our studies was relatively high when compared to other land cover classifications that utilized SAR and RF classifier (e.g., [59]). Our findings are however consistent with similar studies in wetland areas [39,80,81]. Accuracy of the S1 dataset improved significantly, by 8%, when additional Sentinel-1 features were extracted and combined with the original bands for classification. Likewise, the overall accuracy of the S2 dataset improved significantly when combined with additional Sentinel-2 extracted features. The latter observation is contrary to observations made by Tavares et al. [59] who noted decreased accuracy when other optical features were combined with the original bands. The decreased accuracy observed in their case might be due to the presence of irrelevant or redundant features. It is therefore important to optimize datasets by eliminating such redundant features for an improved accuracy in the case of integrating more features.

The relevance of the multi-sensor features for the delineation of land cover components of the tropical landscape using RF classifier is illustrated in Table 7 and Figure 7. Elevation ranked as the most important feature when the landscape was classified with the full dataset (S2+S1+DEM), illustrating the importance of topographical and landform position in peatland occurrence and identification. Peatlands develop under long-term water saturation of the soil and are found in areas where large amounts of water are available or flowing (e.g., rivers, depression). Elevation models are known to be useful for identifying hydrological landscape units [14,36,59]. This was further demonstrated when the individual classes were considered; elevation proved the most important feature for delineating water. We concur with Lidzhegu et al. [38] that topographic information derived from the SRTM can better offer different geomorphologic characteristics which reflect the habitats of different vegetation types and can help in their identification. Despite the Sentinel-1 classifications (S1, S1+) proving relatively inaccurate overall, Sentinel-1 features were consistently rated highly in combined datasets. The addition of Sentinel-2, especially the NDWI feature, clearly leads to a more accurate distinction of peatland and non-peatland classes—this is consistent with observations made by Slagter et al. [80] who also reported the importance of NDWI for wetland delineation. In our study, texture features computed from the Sentinel-1 image were of relatively low importance and were often removed as redundant features. This could be because of the presence of other features such as the original Sentinel-1 bands and the standard deviations of the VV, VH and NDVI bands which played similar roles thus rendering the Sentinel-1 GLCM texture features less useful. 

The sensitivity of land cover types to classification features was demonstrated in Table 7. For example, when discriminating tropical peatland classes—mangrove, palm and mixed swamp—Sentinel-1’s VV, VH and the standard deviation of the VH bands acquired higher importance scores than the optical and elevation features considered. This may be because microwaves from Sentinel-1 penetrate forests and interact with different parts of trees to produce substantial volume scattering. As the importance of VH and VV was high, it is likely that volume scattering (especially for VH in medium- and high-vegetated peatlands), double-bounce scattering (especially for VV in low- and medium-vegetated peatlands) and specular reflection (especially for VV in non- and low-vegetated peatlands) contributed to accurate classification of peatland classes [80]. When distinguishing between rubber and oil palm, elevation was among the most important features. Elevation was also the most useful feature for separating natural forest from peatland forest (e.g., mixed swamp, palm swamp). This demonstrates that the right combination of multi-sensor features is important for the discrimination of diverse land cover types as they maximize the complementarity of the optical spectral sensitivity and the radar structural/geometric characteristics. 

Our analysis estimates Ghana’s Greater Amanzule peatland at 60,187 ha, comprising mangrove, mixed swamp, palm swamp and bog plain. This is a relatively large tropical peatland with no formal/legal protection [15,16]. To date, community and NGO efforts to manage sections of the peatland have tended to focus on mangrove forest. The results (Table 8) however show that mangrove occupies the smallest area of the peatland classes on the landscape. This underlines the need to broaden the scope of management foci to include the other predominant peat classes. This is important because peatlands function as hydrological landscape units; the hydrological connectedness means damage to one part can have wide-reaching consequences on the whole system. Conservation efforts that concentrate on only certain parts of a peatland unit may therefore allow peatland degradation due to activities outside the conserved zones, thus reducing the effectiveness of that conserved area in achieving its conservation goals. An option to manage the unit would be to extend the boundaries of existing conserved areas to the hydrological boundaries of the peatlands that they encompass. Our analysis suggests that plantation development may be a major threat to the Greater Amanzule peatland. This development bears close resemblance to experiences with peatland landscapes in Southeast Asia, where plantation development, predominantly rubber and oil palm, has been reported as the major threat to peatlands [12,74,75,76,82,83]. This points to the need to manage plantation development on the Greater Amazule landscape to ensure its sustainability. Available reports though suggest that plantation developers have not been actively engaged in the current efforts by NGOs and communities in their informal management of the landscape [15,16]. At present, the Greater Amanzule peatland is relatively intact, hence the need to fully investigate the threat and conservation priorities of the landscape to aid management decisions. Research is also needed to evaluate the carbon stock of the Greater Amanzule peatland to complement the work done by Ajonina et al. [46] and Asante and Jengre [45] who investigated carbon stock in sections of the peatland. This will aid understanding of the potential of the landscape to attract climate change mitigation funding to the benefit of fringe communities. More broadly assessing the current and future potential of the Greater Amanzule peatland to supply ecosystem services will help underline its importance and motivate public protection and conservation of its unique ecosystem functions and services [84]. The results from this study can be used as a baseline for onward analysis of land cover change to understand the impact of plantation development and to simulate future land uses. For instance, the distribution of coconut plantations in the study area is reported to have been affected by a coconut disease that killed most coconuts in the eastern part of the landscape. Time series analysis may again help to quantify the impact of the disease, and subsequently help to prepare for similar situations in the future [16]. Finally, even though our proposed framework is implemented on a relatively large area (e.g., when compared to [74,83]), we still recommend its application on an even larger area to better understand how the model will perform on more complex and heterogenous landscapes.

## 5. Conclusions

Tropical peatlands are highly valuable ecosystems and under great pressure from anthropogenic land use activities. Accurate information on the distribution and extent of peatland is therefore important for the sustainable management of remnant tropical peatlands, especially in Africa where information on peatland is generally scarce. This study presents the first attempt to define the extent and distribution of Ghana’s Greater Amanzule tropical peatland using freely available multi-source satellite imagery and a robust machine learning approach. We demonstrated the successful application of integrated optical, radar and elevation data for mapping tropical peatland and demonstrated how carefully selected data features maximise peatland classification accuracy. We further analysed the sensitivity of land cover types to multi-source data features. From our analysis, integrated Sentinel-2, Sentinel-1 and SRTM features (S2+S1+DEM) yielded the highest classification accuracy, significantly outperforming five other dataset combinations. Analysis of the sensitivity of land cover classes to multi-source features showed that elevation and radar extracted features, particularly VV and VH, were important predictors for tropical peatland delineation. We estimate the Greater Amanzule tropical peatland covers 60,187 ha. The proposed methodological framework provides a reliable and robust workflow for accurate landscape-scale monitoring of tropical peatlands. Additionally, our findings provide timely baseline information critical for supporting the development of sustainable and adaptive management strategies for conservation priorities, monitoring deforestation and forest degradation, quantifying the carbon stock of the Greater Amanzule landscape and supporting restoration projects. 

## Figures and Tables

**Figure 1 sensors-21-03399-f001:**
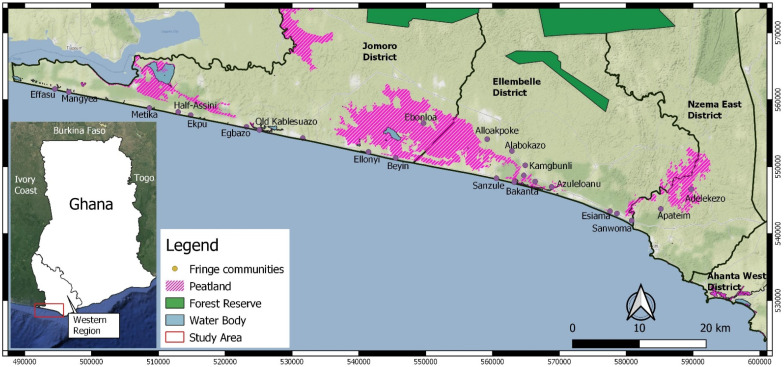
The Greater Amanzule landscape showing identified patchy peatlands and communities fringing the wetland resources. Peatland information was obtained from Hen Mpoano’s data repository and is based on participatory GIS and ground truthing approach.

**Figure 2 sensors-21-03399-f002:**
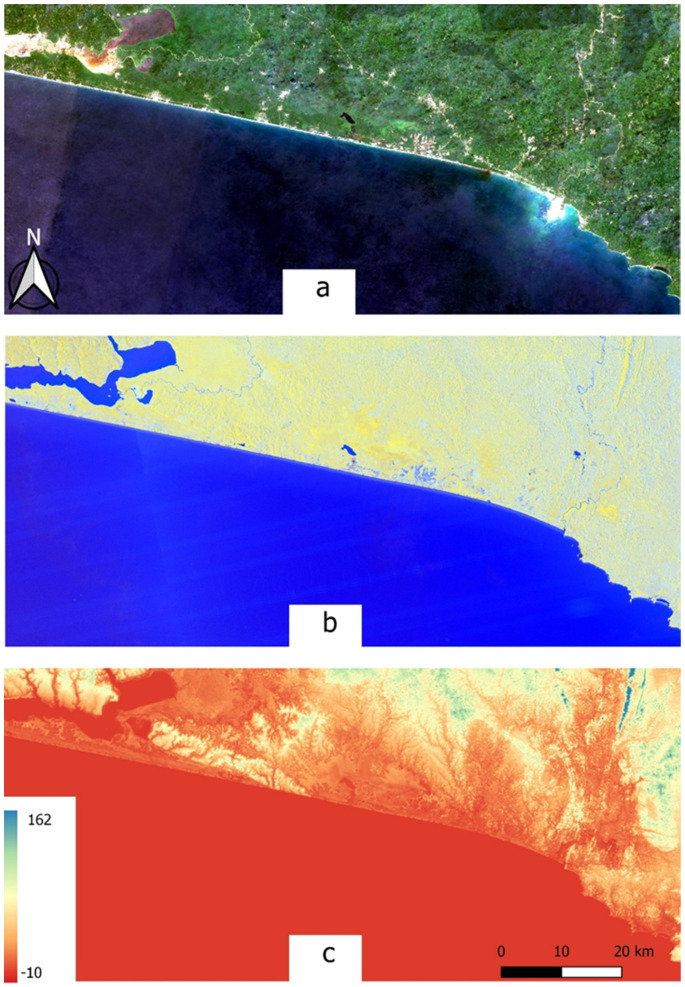
Satellite image data of the study area: (**a**) Sentinel-2 true colour composite, (**b**) Sentinel-1 dual-polarization and (**c**) SRTM DEM showing estimated elevation in metres above sea level.

**Figure 3 sensors-21-03399-f003:**
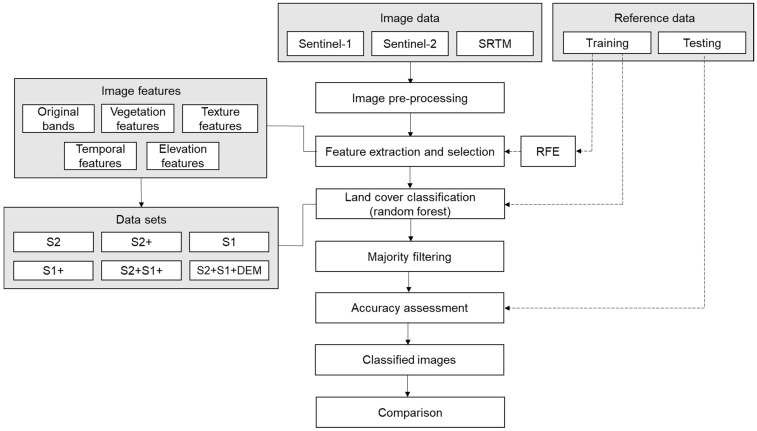
Process flowchart for land cover classification of Greater Amanzule using individual and integrated Sentinel-2, Sentinel-1 and SRTM datasets.

**Figure 4 sensors-21-03399-f004:**
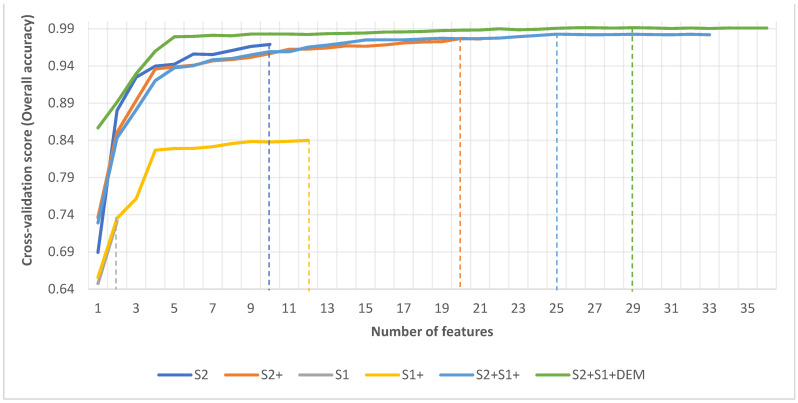
Optimal Features selected for the classification of each datasets using RFE algorithm.

**Figure 5 sensors-21-03399-f005:**
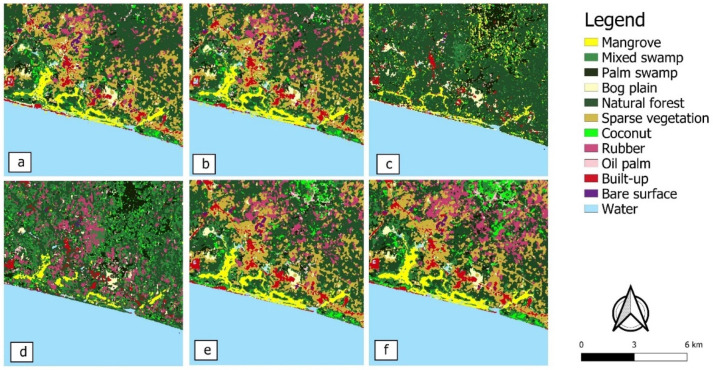
Land cover classification results of a small part of the study area (zoomed in for ease of viewing) using the (**a**) S2, (**b**) S2+, (**c**) S1, (**d**) S1+, (**e**) S2+S1+ and (**f**) S2+S1+DEM datasets.

**Figure 6 sensors-21-03399-f006:**
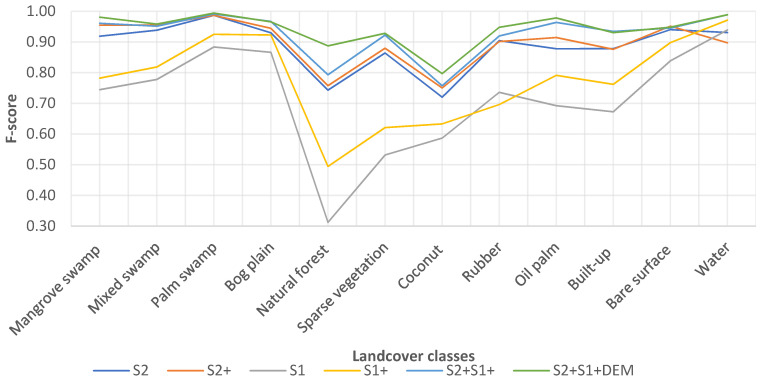
F-score of landcover classes from the classification results of various datasets.

**Figure 7 sensors-21-03399-f007:**
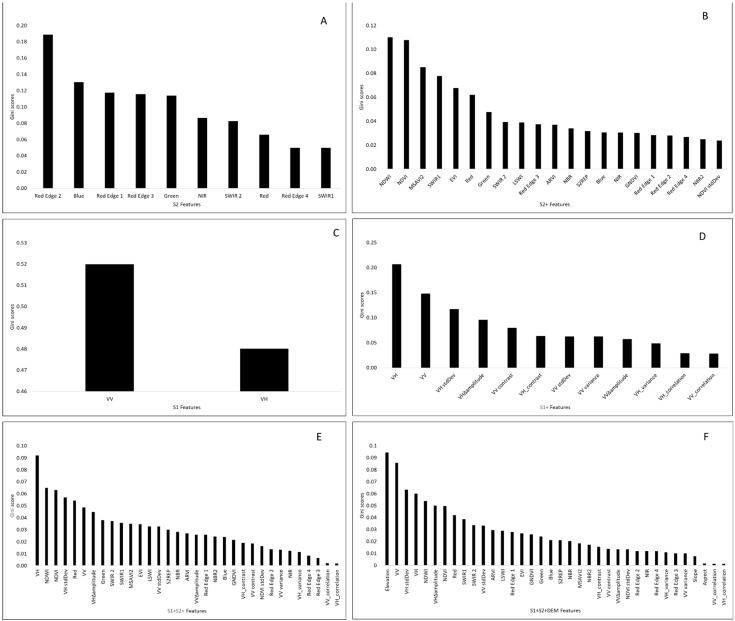
Important predictor variables from RF classification results: (**A**) S2, (**B**) S2+, (**C**) S1, (**D**) S1+, (**E**) S2+S1+ and (**F**) S2+S1+DEM.

**Figure 8 sensors-21-03399-f008:**
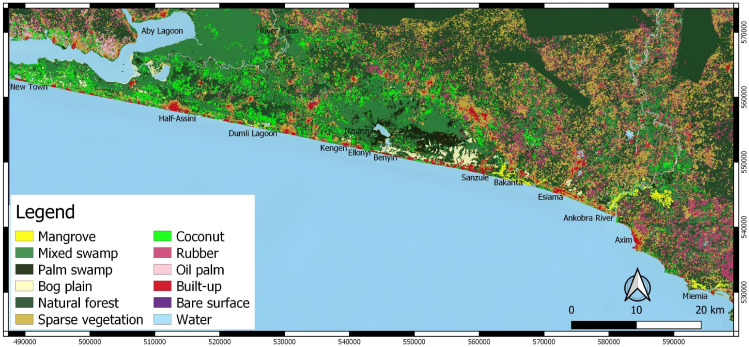
Land cover classification of the Greater Amanzule peatland based on the S2+S1+DEM dataset.

**Table 1 sensors-21-03399-t001:** Satellite remote sensing data used for Greater Amanzule land cover classification.

Data	Period	Bands	Number of Images
Sentinel -1 SAR GRD: C-band Synthetic Aperture Radar Ground Range Detected, log scaling	2 January 2019–31 December 2019	VV, VH	63
Sentinel-2 MSI: Multispectral Instrument, Level-1C	1 January 2019–28 December 2019	B1, B2, B3, B4, B5, B6, B7, B8, B8A, B9, B10, B11, B12 (B1 and B10 were only used for cloud detection)	366
Shuttle Radar Topography Mission (SRTM) digital elevation dataset	11 February 2000–12 February 2000	Elevation	1

**Table 2 sensors-21-03399-t002:** Implemented land cover classes and the associated reference data. Training samples are expressed as total area of polygons and number of pixels extracted from these polygons, and test samples represent number of points.

General Class	Land Cover Class	Class Description	Training Samples		Test Samples
			Area of Polygons (ha)	Number of Extracted Pixels	
Peatland	Mangrove swamp	Mangrove cover along coastal areas	24.1	1862	265
	Mixed swamp	Permanent and regularly flooded broadleaved trees and palm (Raphia sp.)	241.3	5727	589
	Palm swamp	Permanent and regularly flooded areas of palm (predominantly Raphia sp.)	50.9	1312	334
	Bog plain	Areas dominated by permanent and regularly flooded areas of grasses	136.98	3479	269
Forest	Natural forest	Closed broadleaved evergreen forest with trees from medium to large sizes	331.3	11,738	115
Sparse	Sparse vegetation	Areas of sparse and/or stunted plant growth including other agricultural lands (i.e., young plantation trees, rainfed croplands)	5.8	301	242
Plantation	Coconut	Plantation of mature coconut trees	39.6	1350	282
	Rubber	Plantation of mature rubber trees	49.3	1734	228
	Oil palm	Plantation of mature oil palm trees	26	656	70
Artificial and bare areas	Built-up	Developed land such as buildings, asphalt roads and concrete surfaces, human settlements, industrial facilities	41.7	1215	258
	Bare surface	Areas of exposed soil or ground/open areas devoid of trees, grass or other vegetation; often comprising land cleared for development	3.5	101	172
Hydrology	Water	Water bodies such as rivers, canals, lakes and sea	709	18,195	87
Total			1659.48	47,670	2911

**Table 3 sensors-21-03399-t003:** Features considered for land cover classification.

Dataset	Source	Index	Number of Features	References
S2	Sentinel-2 bands	Blue, Red, Green, NIR, SWIR1, SWIR2, Red Edge1, Red Edge2, Red Edge3, Red Edge4	10	[20,21,22,23,24,25,26,52]
S2+	Sentinel-2 bands plus extracted vegetation index and texture feature	Blue, Red, Green, NIR, SWIR1, SWIR2, Red Edge1, Red Edge2, Red Edge3, Red Edge4, NDVI, GNDVI, LSWI, S2REP, NDWI, NBR, NBR2, EVI, ARVI, MSAVI2, NDVI_stdDev (standard deviation–texture)	21	[41,44,48,53,54]
S1	Sentinel-1 bands	VH, VV	2	[32,39,49,53]
S1+	Sentinel-1 bands plus extracted texture and temporal features	VH, VV, VV_correlation, VV_variance, VV_contrast, VH_correlation, VH_variance, VH_contrast, VV_stdDev, VH_stdDev, VV_Δamplitude_, VH_Δamplitude_	12	[32,39,49,53]
S2+S1+	Sentinel-2 and Sentinel-1 bands, plus extracted features	Blue, Red, Green, NIR, SWIR1, SWIR2, Red Edge1, Red Edge2, Red Edge3, Red Edge4, NDVI, GNDVI, LSWI, S2REP, NDWI, NBR, NBR2, EVI, ARVI, MSAVI2, NDVI_stdDev, VH, VV, VV_correlation, VV_variance, VV_contrast, VH_correlation, VH_variance, VH_contrast, VV_stdDev, VH_stdDev, VV_Δamplitude_, VH_Δamplitude_	33	[41,44,48,53,54]
S2+S1+DEM	Sentinel-2 and Sentinel-1 bands, plus extracted features, plus SRTM elevation features	Blue, Red, Green, NIR, SWIR1, SWIR2, Red Edge 1, Red Edge 2, Red Edge 3, Red Edge 4, NDVI, GNDVI, LSWI, S2REP, NDWI, NBR, NBR2, EVI, ARVI, MSAVI2, NDVI_stdDev, VH, VV, VV_correlation, VV_variance, VV_contrast, VH_correlation, VH_variance, VH_contrast, VV_stdDev, VH_stdDev, VV_Δamplitude_, VH_Δamplitude,_ Elevation, Slope, Aspect	36	[36,37,38]

**Table 4 sensors-21-03399-t004:** Image features retained for land cover classification.

Datasets	Index
S2	Blue, Red, Green, NIR, SWIR1, SWIR2, Red Edge 1, Red Edge 2, Red Edge 3, Red Edge 4
S2+	Blue, Red, Green, NIR, SWIR1, SWIR2, Red Edge 1, Red Edge 3, Red Edge 4, NDVI, GNDVI, LSWI, S2REP, NDWI, NBR, NBR2, EVI, ARVI, MSAVI2, NDVI_stdDev
S1	VH, VV
S1+	VH, VV, VV_correlation, VV_variance, VV_contrast, VH_correlation, VH_variance, VH_contrast, VV_stdDev (standard deviation), VH_stdDev, VV_Δamplitude_, VH_Δamplitude_
S2+S1+	Blue, Red, Green, SWIR1, SWIR2, Red Edge 1, NDVI, GNDVI, LSWI, S2REP, NDWI, NBR, NBR2, EVI, ARVI, MSAVI2, NDVI_stdDev, VH, VV, VV_contrast, VH_var, VH_contrast, VV_stdDev, VH_stdDev, VH_Δamplitude_
S2+S1+DEM	Blue, Red, Green, NIR, SWIR1, SWIR2, Red Edge 1, Red Edge 4, NDVI, GNDVI, LSWI, S2REP, NDWI, NBR, NBR2, EVI, ARVI, MSAVI2, NDVI_stdDev, VH, VV, VV_var, VH_var, VH_contrast, VV_stdDev, VH_stdDev, VV_Δamplitude_, VH_Δamplitude,_ Elevation

**Table 5 sensors-21-03399-t005:** McNemar’s chi-squared test score (z) of data pairs. Values in parenthesis represent *p*-value. Data pairs that show statistically significant difference (*p* ≤ 0.05) and the best overall accuracy (OA) are in bold.

	Datasets					
	S2	S2+	S1	S1+	S2+S1+	S2+S1+DEM
S2		8.4767	1387.7	288.21	53.125	72.755
		**(0.0036)**	**(0.0000)**	**(0.0000)**	**(0.0000)**	**(0.0000)**
S2+			1429.7	334.89	29.009	47.617
			**(0.0000)**	**(0.0000)**	**(0.0000)**	**(0.0000)**
S1				987.85	1516.9	1541.4
				**(0.0000)**	**(0.0000)**	**(0.0000)**
S1+					426.97	440.41
					**(0.0000)**	**(0.0000)**
S2+S1+						4.0635
						**(0.0438)**
OA	89.83	91.03	70.95	78.02	92.88	**94.30**

**Table 6 sensors-21-03399-t006:** Producer’s accuracy (PA) and user’s accuracy (UA) of land cover classes for all datasets, and the difference in UA and PA between S2+S1+DEM and each of the other datasets. Negative values indicate lower UA and PA for S2+S1+DEM compared to the other datasets. The best UA and PA values for each class are in bold.

Land Cover Classes	S2		S2+		S1		S1+		S2+S1+		S2+S1+DEM	
	UA	PA	UA	PA	UA	PA	UA	PA	UA	PA	UA	PA
Mangrove	99.6	85.3	**100.0**	91.3	74.2	74.7	85.0	72.5	**100.0**	92.5	**100.0**	**96.2**
Mixed swamp	88.8	99.5	91.9	99.5	82.4	73.7	78.7	85.0	91.1	99.5	**92.3**	**99.7**
Palm swamp	98.2	99.1	98.5	98.8	85.0	91.9	89.4	95.8	98.8	**99.4**	**99.4**	**99.4**
Bog plain	88.3	**98.1**	91.3	97.8	78.0	97.4	87.4	97.8	**96.0**	97.4	95.0	**98.1**
Natural forest	60.8	95.7	62.4	96.5	18.9	90.4	33.2	96.5	66.5	98.3	**80.3**	**99.1**
Sparse vegetation	92.1	81.4	93.5	83.1	98.9	36.4	**100.0**	45.0	93.6	**90.9**	94.8	**90.9**
Coconut	88.6	60.6	88.8	64.9	78.6	46.8	83.2	51.1	90.2	65.3	**90.9**	**70.9**
Rubber	94.3	86.8	92.2	88.2	87.4	63.6	64.0	76.3	94.0	89.9	**94.4**	**95.2**
Oil palm	88.4	87.1	91.4	91.4	75.0	64.3	79.7	78.6	98.5	94.3	**100.0**	**95.7**
Built-up	87.6	88.0	87.6	87.6	81.5	57.3	88.0	67.2	**90.6**	**96.5**	90.5	95.7
Bare surface	96.9	**91.3**	**100.0**	90.7	97.7	73.4	96.7	83.8	99.4	90.1	**100.0**	90.1
Water	87.0	**100.0**	81.3	**100.0**	**100.0**	88.5	97.7	96.6	97.8	**100.0**	97.8	**100.0**
**Difference (S2+S1+DEM—other datasets)**												
Mangrove	0.4	10.9	0.0	4.9	25.8	21.5	15.0	23.7	0.0	3.7		
Mixed swamp	3.5	0.2	0.4	0.2	9.9	26.0	13.6	14.7	1.2	0.2		
Palm swamp	1.2	0.3	0.9	0.6	14.4	7.5	10.0	3.6	0.6	0.0		
Bog plain	6.7	0.0	3.7	0.3	17.0	0.7	7.6	0.3	−1.0	0.7		
Natural forest	19.5	3.4	17.9	2.6	61.4	8.7	47.1	2.6	13.8	0.8		
Sparse vegetation	2.7	9.5	1.3	7.8	−4.1	54.5	−5.2	45.9	1.2	0.0		
Coconut	2.3	10.3	2.1	6.0	12.3	24.1	7.7	19.8	0.7	5.6		
Rubber	0.1	8.4	2.2	7.0	7.0	31.6	30.4	18.9	0.4	5.3		
Oil palm	11.6	8.6	8.6	4.3	25.0	31.4	20.3	17.1	1.5	1.4		
Built-up	2.9	7.7	2.9	8.1	9.0	38.4	2.5	28.5	−0.1	−0.8		
Bare surface	3.1	−1.2	0.0	−0.6	2.3	16.7	3.3	6.3	0.6	0.0		
Water	10.8	0.0	16.5	0.0	−2.2	11.5	0.1	3.4	0.0	0.0		

**Table 7 sensors-21-03399-t007:** S2+S1+DEM feature importance for discriminating various land cover types (a, b, c, d and e in shaded cells represent the five most important features, respectively).

Classification Features	Land Cover Classes											
	Mangrove	Mixed Swamp	Palm-Swamp	Bog Plain	Natural Forest	Sparse Vegetation	Rubber	Coconut	Oil Palm	Built-Up	Bare Surface	Water
Blue	0.001	0.001	0.005	0.025	0.010	0.004	0.003	0.003	0.005	0.097	0.073	0.005
Green	0.017	0.018	0.034	0.050 e	0.001	0.032	0.011	0.015	0.019	0.136 e	0.137 e	0.005
Red	0.010	0.014	0.027	0.109 a	0.009	0.032	0.006	0.012	0.015	0.282 a	0.416 a	0.033
Red Edge 1	0.012	0.011	0.018	0.027	0.007	0.019	0.009	0.011	0.013	0.048	0.077	0.001
Red Edge 2	0.010	0.009	0.013	0.008	0.012	0.016	0.013	0.012	0.014	0.009	0.020	0.009
Red Edge 3	0.009	0.009	0.013	0.007	0.013	0.017	0.015	0.013	0.016	0.007	0.017	0.010
NIR	0.024	0.024	0.033	0.020	0.036	0.045	0.038	0.033	0.041	0.018	0.039	0.027
Red Edge 4	0.008	0.009	0.011	0.007	0.012	0.015	0.013	0.012	0.014	0.006	0.012	0.009
SWIR 1	0.008	0.016	0.030	0.049	0.023	0.045	0.033	0.021	0.09	0.075	0.111	0.024
SWIR 2	0.003	0.018	0.035	0.102 b	0.024	0.066 c	0.039	0.021	0.032	0.240 b	0.251 b	0.031
NDVI	0.038	0.037	0.039 d	0.020	0.046	0.043	0.045	0.042 d	0.045	0.003	0.001	0.034
GNDVI	0.010	0.006	0.009	0.064 d	0.014	0.001	0.006	0.005	0.006	0.106	0.212 c	0.041 d
NDWI	0.038	0.038 e	0.041 c	0.030	0.048	0.046 e	0.048 d	0.044 c	0.047 d	0.013	0.027	0.030
EVI	0.027	0.027	0.036 e	0.011	0.044	0.050 d	0.045 e	0.039 e	0.047 e	0.014	0.005	0.036 e
MSAVI2	0.022	0.021	0.028	0.007	0.035	0.036	0.035	0.030	0.036	0.011	0.007	0.027
LSWI	0.025	0.005	0.008	0.099 c	0.014	0.014	0.003	0.011	0.0125	0.189 c	0.161 d	0.026
ARVI	0.000	0.000	0.000	0.000	0.000	0.000	0.000	0.000	0.000	0.000	0.000	0.000
NBR	0.029	0.018	0.015	0.049	0.024	0.008	0.017	0.022	0.024	0.165 d	0.094	0.007
NBR 2	0.017	0.015	0.017	0.008	0.017	0.016	0.018	0.018	0.018	0.006	0.012	0.010
S2REP	0.000	0.000	0.000	0.000	0.000	0.000	0.000	0.000	0.000	0.000	0.000	0.000
NDVI_stdDev	0.027	0.001	0.002	0.009	0.001	0.006	0.000	0.002	0.001	0.019	0.055	0.005
VH	0.106 a	0.106 a	0.092 a	0.045	0.101 b	0.093 a	0.097a	0.097 a	0.085 a	0.083	0.022	0.042 c
VV	0.093 b	0.089 b	0.083 b	0.016	0.082 c	0.070 b	0.074 c	0.082 b	0.079 b	0.072	0.018	0.054 b
VH_stdDev	0.051 c	0.058 c	0.027	0.008	0.058 c	0.038	0.039	0.038	0.024	0.044	0.018	0.032
VV_stdDev	0.038	0.036	0.024	0.002	0.034	0.017	0.020	0.026	0.022	0.059	0.001	0.021
VV_variance	0.001	0.001	0.002	0.001	0.008	0.000	0.001	0.001	0.002	0.029	0.011	0.002
VV_contrast	0.002	0.001	0.003	0.001	0.011	0.000	0.001	0.002	0.003	0.044	0.015	0.003
VV correlation	0.000	0.000	0.000	0.000	0.000	0.000	0.001	0.000	0.000	0.000	0.001	0.000
VH variance	0.000	0.000	0.003	0.000	0.007	0.002	0.000	0.002	0.003	0.018	0.026	0.002
VH contrast	0.000	0.000	0.006	0.001	0.012	0.004	0.001	0.004	0.005	0.036	0.046	0.004
VH correlation	0.000	0.000	0.000	0.000	0.000	0.000	0.001	0.000	0.000	0.000	0.001	0.000
VV_Δamplitude_	0.021	0.019	0.014	0.001	0.019 e	0.010	0.011	0.015	0.012	0.030	0.004	0.011
VH_Δamplitude_	0.048 d	0.054 d	0.025	0.002	0.053	0.036	0.036	0.036	0.022	0.036	0.017	0.030
Aspect	0.001	0.001	0.001	0.001	0.001	0.001	0.001	0.001	0.001	0.001	0.001	0.001
slope	0.0002	0.0006	0.001	0.002	0.007	0.003	0.008	0.002	0.000	0.001	0.005	0.004
Elevation	0.041 e	0.016	0.014	0.046	0.234 a	0.046	0.087 b	0.030	0.060 c	0.039	0.036	0.066 a

**Table 8 sensors-21-03399-t008:** Land cover class areas in Greater Amanzule classification.

General Class	Classes	Area (ha)	Percentage of Study Area	Area of General Class (ha)	General Class Percentage of Study Area
Peatland	Mangrove swamp	1633.78	0.28	60,187.04	10.29
Mixed swamp	48,851.29	8.35
Palm swamp	5143.97	0.88
Bog plain	4558.00	0.78
Forest	Natural forest	102,728.14	17.57	102,728.14	17.57
Sparse	Sparse vegetation	41,856.35	7.16	41,856.35	7.16
Plantation	Coconut	18,109.00	3.10	50,713.28	8.67
Rubber	29,998.06	5.13
Oil palm	2606.22	0.45
Artificial and bare areas	Built-up	5273.31	0.90	5363.56	0.92
Bare surface	90.25	0.02
Hydrology	Water	323,878.13	55.39	323,878.13	55.39
Total	584,726.50		584,726.50

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
