# Peer review of "Testing the Contribution of Multi-Source Remote Sensing Features for Random Forest Classification of the Greater Amanzule Tropical Peatland"

_sensors, 2021, doi:10.3390/s21103399_

Round 1

Reviewer 1 Report

The submited paper is interesting and clearly written. It is overall a good work. I only have two concerns.

  1. The study area is not very large and seems environmentally homogeneous. My own experience indicates that the larger the study area, and the environmental heterogeneity, the most difficult is to train a good classification model. This issue is especially serious when topographical variables are added as predictors. Do you think your results would be so good in a larger area? Of course thsis is not at all a demerit of your work, nut I thonk it is worth to be mentioned in the discussion.

2. A more serious concern is that you don't seem to have a validation set. The test set should not be used to optimize the hyperparameters (ntree, mtry or feature sets) of a predictive model or it will not be useful to provide a proper error estimation as the model has been previously optimized using it. So, the most advisale option is to have 3 sets (training, validation and set).  If you have optimized your model using the test set, I think you shouls call it validation test and produce a new test set to obtain the final accuracy measurement.

Reviewer 2 Report

Overview:

This study demonstrates a land cover classification methodology focused on tropical peatland areas. The various experiments are implemented in Google Earth Engine and based on a random forest algorithm and using S1, S2 and DEM data. Six different feature combinations (consisting of bands, indices, temporal features and texture features) were applied and assessed in a small study area and the best performing combination was used to apply the methodology on a broader area.

The study is well written and structured. The topic, the methods and the results are clearly presented, in general.

General comments and suggestions for authors:

Since you are using DEM data in your study, it is important to provide information on the topography of the study area in section 2.1.

Please provide the preprocessing level of the S2 data in the text as well (L1C) besides Table 1. Why weren’t the L2A level S-2 products used instead? Why weren’t any atmospheric corrections applied to the S2 data?

What is the red box in the figure 1?  In combination with figure 4 it gives the impression of a test area, where the methodology was tested and a broader area, where the methodology was applied. If this is the case, name your two study areas (eg “test area” and “study area”) so it can be clear from the beginning of the paper. Also, clearly name the areas in the legend of Figure 1. In regard to this, it must also be clearly stated in the text, from which study area the reference samples came from.

I believe Table 7 illegible to the reader. Perhaps this table could be turned into a graph, similar to Figure 6, with a separate graph for each class. Or maybe colors could be added to highlight the large values in the table.

Comments by line:

141                         What is the red box in the figure? Please delineate the exact study area boundaries or if the red box represents them, add it to the legend of the map accordingly.

168                         Why is “undefined peatland boundaries” considered a threat? This statement is too general.

195                         The creation of only one composite S2 image for the whole year would produce errors due to the changes in seasons. Why was there only one composite created instead of two (eg. one for the wet season and one for dry season)?

196                         What benefit did the mean composite of the 40-60 percentile range offer over the mean composite of the full range of values?

215                         Were any GIS techniques applied to achieve uniformity of reference data? Was any minimum distance used?

227                         Since training samples were collected as polygons and test samples in the form of points, in what unit are the reference samples presented in the table? Pixels, polygons?

430-432                It is evident from the accuracy metrics in table 6 that the inclusion of the DEM-derived information improved the discrimination of high-vegetation classes (forest, coconut, rubber, palm, sparse) as well as built-up areas. This indicates that the DEM information may work more as a DSM information rather than true topography information. This should be highlighted in the discussion section.

430                         Important variables may occur in RF because a feature (eg elevation) might have considerable differences in its value range. Were all the features rescaled to the same range before performing variable importance?

435-437                It might be better for this sentence to be moved in the discussion section.

447-449                It might be better for this sentence to be moved in the discussion section.

451-454                It might be better for this sentence to be moved in the discussion section.

456-457                It might be better for this sentence to be moved in the discussion section.

468-474                It might be better for this sentence to be moved in the discussion section.

477-491                It might be better for this sentence to be moved in the discussion section.

Reviewer 3 Report

Testing the contribution of multi-source remote sensing fea- tures for random forest classification of the Greater Amanzule tropical peatland

This revised manuscript evaluates the contribution of multi-source imagery (i.e., radar, optical, and elevation dana) for tropical peatland mapping. Furthermore, different classification scenarios were investigated using Random Forest (RF) algorithm and assessment of land cover classes to image features was done.

This manuscript needs a major revision before it is considered for potential publication in this journal. My major concerns are listed as follows:

  • In Abstract term „Random forest-recursive feature elimination“ is mentioned. Recursive feature elimination (RFE) is an embedded type of feature selection algorithm, but as mentioned in Abstract, reader might think it is a built-in Random Forest method. Also, LN 322 refers recursive feature elimination as REF, and later in manuscript RFE is used.
  • Table 2 and further in manuscript, class Natural forest is mentioned, whereas later the authors mention and refer to the Mixed forest. This needs to be adjusted
  • Table 3, S2+ - which texture features were used in this classification scenario?
  • LN 332: please describe in detail about Gini index and mention which another measure for feature importance scores exists
  • Eq 14: F-score equation is presented as f1 and different abbreviation is used later
  • Section 2.3. mentions training and testing data, and Section 2.5. addressed as „… and evaluation“ the reader does not receive the information about validation protocol (sampling design, how the split between train and validation data is made (and the purpose of using polygon and point data for classification).

Please take a look at these two papers :

"Key issues in rigorous accuracy assessment of land cover products" by Stehman & Foody (2019).

"Good practices for estimating area and assessing accuracy of land change" by Olofsson & al (2014).

  • Figure 3: caption on y-axis – what does it mean „no of correct classification“?
  • Table 5 and 6: a detailed preview of the results has been made, but it is pretty difficult to derive some conclusions from them. I suggest to put OA, UA and PA in the range from 0-100, and in Table 6 perhaps use bold and italic values to emphasise best UA/PA value of each class
  • LN 464: I do not think that the extent and distribution is shown on Figure 6
  • Figure 7 – in what coordinate system are expressed x and y axis? Ranges over longitude/latitude cover a wide area according to the range of each axis

I think that the research design of this manuscript is good, but still some additional changes need to be made, in order to be published in this journal.

Round 2

Reviewer 1 Report

Ok. Good work.

Best wishes

Author Response

Thanks for your positive review and contribution towards the manuscript.

Reviewer 3 Report

Testing the contribution of multi-source remote sensing features for random forest classification of the Greater Amanzule tropical peatland

The aforementioned manuscript has improved from version_01 to version_02. However, I still have some concerns, and they are listed, as follow:

  • Ad 1: the authors mention that the term random forest has been removed as recommended. However, still RF-RFE term exists in LN 268, 314, 331 and my opinion is that aforementioned term still makes confusion
  • Ad 3: For me, it is still not clear how/or which texture has been calculated from the standard deviation of the NDVI
  • Ad 7: y-axis, still the accuracy metric is not mentioned (I suppose that it is Overall accuracy)
  • Figure 8 – although it was the easiest way, I think that the spatial extent of the land cover classification map should be presented.

Overall, the manuscript has improved between the two versions, but still in some parts of the manuscript some additional changes need to be made.

Author Response

  • Ad 1: the authors mention that the term random forest has been removed as recommended. However, still RF-RFE term exists in LN 268, 314, 331 and my opinion is that aforementioned term still makes confusion

RESPONSE: Lines 268, 314 and 331 have been edited to remove the term random forest (RF) as recommended.

  • Ad 3: For me, it is still not clear how/or which texture has been calculated from the standard deviation of the NDVI.

 RESPONSE: The following clarification has been added to the text: “Standard deviation metrics were computed on Sentinel-2 based NDVI and Sentinel-1’s VH and VV bands using a 5 x 5 pixels moving window [41]. Thus, for each central pixel in the 5 x 5 window, the standard deviation of the 25 pixels (in the window) was calculated and the value applied to the corresponding (central) pixel in the output texture image.”.

  • Ad 7: y-axis, still the accuracy metric is not mentioned (I suppose that it is Overall accuracy)

 RESPONSE: Overall accuracy has been added to the y-axis.

  • Figure 8 – although it was the easiest way, I think that the spatial extent of the land cover classification map should be presented.

 RESPONSE: We are not clear on exactly what you mean by the ‘spatial extent’ of the map. We have interpreted this to mean the map coordinates along the borders, and we have added these back in.